

# The effect of perceived stress for work engagement in volunteers during the COVID-19 pandemic: the mediating role of psychological resilience and age differences

Yatong Li[1,2,*], Pei Xie[1,*], Liang He[1], Xiaolong Fu[1], Xiaobing Ding[1], Mary C. Jobe[3] and Md Zahir Ahmed[4]

[1] School of Psychology, The Northwest Normal University, Lanzhou, China
[2] Longnan Teachers College, Chengxian, China
[3] Department of Psychological and Brain Sciences, George Washington University, Washington D.C., United States of America
[4] School of Psychology, Zhejiang Normal University, Jinhua, Zhejiang, China
[*] These authors contributed equally to this work.

Corresponding authors
Yatong Li, liyatongxbsd@outlook.com
Md Zahir Ahmed, ahmedzahirdu@gmail.com

## ABSTRACT

Volunteers played an important role throughout the COVID-19 pandemic. This study investigated the characteristics of perceived stress, psychological resilience and work engagement among 910 Chinese volunteers of different ages in the first month of pandemic in Gansu province, China. The present study tested the correlations between perceived stress and work engagement, the mediating role of psychological resilience in the relationship and the differences among age groups. The results of this study showed that work engagement and psychological resilience increased with the age of the volunteers. Work engagement and resilience levels were higher in middle adulthood than in early adulthood. As predicted, perceived stress negatively predicted work engagement. A mediation analysis showed that psychological resilience partially explained the correlations between perceived stress and work engagement. Specifically, the mediating effect of psychological resilience in early adulthood was significant, but not in middle adulthood. Overall, this study demonstrates that work engagement increased with age and was negatively predicted by perceived stress, showing these factors were important for volunteers' work during COVID-19. Further, for those in early adulthood, psychological resilience mediated this relationship—highlighting another age difference among volunteers during COVID-19.

## INTRODUCTION

The COVID-19 pandemic poses a severe challenge to governments worldwide and is a considerable threat to public health. While the mass vaccination campaign was launched in early 2021, the pandemic continued worldwide. When COVID-19 spread in China, volunteer teams were set up quickly to carry out all kinds of services to prevent and control

COVID-19's spread. When stay-at-home orders were in place, volunteers made sacrifices to their personal safety to ensure their communities did not spread COVID-19. With the contributions of these volunteers being central to uphold all aspects of community COVID-19 prevention and control, the extent to which they are engaging in their work has become a topic of interest. Work engagement can be regarded as person's active participation in work, which is predictive of one's work efficiency and attributes to a positive, reciprocal psychological state distinguished by vitality, dedication and concentration (*Schaufeli, Bakker & Salanova, 2006*). For volunteers, they must have high work engagement—in this case, a strong guarantee for the implementation of COVID-19 prevention. Additionally, stress and psychological resilience have an impact on work engagement (*Schaufeli & Bakker, 2004*; *Zhang, Zhu & Yu, 2022*). Therefore, there is a considerable level of implication to explore the mechanism of stress and psychological resilience on volunteer work engagement to understand how each contributes to volunteer's level of dedication in controlling and preventing the spread of COVID-19.

## Perceived stress and work engagement

It is widely acknowledged that shifts throughout a pandemic might function as major aggravation that persuades an elevated amount of stress. Whenever one experiences the constraints of such stress surpassing their coping mechanism or adaptabilities, it might trigger elevated negative emotions, and physical sensations, resulting in a burdensome disruption in their work engagement (*Lazarus & Folkman, 1984*; *Liu et al., 2019*). Perceived stress, rather than the objective stressful event, is more relevant to focus on; objectively stressful events may occur, but how they are perceived and affect individuals may vary. Perceived stress is defined as a personal psychological response and subjective feelings when confronted with negative situations (*Zheng et al., 2019*). Negative emotions are closely correlated with perceived stress (*Schiffrin & Nelson, 2010*; *Spada et al., 2008*). Existing studies illustrate that emotion is the key determinant of work engagement, especially negative emotions (*Firdaus, 2019*; *Liu et al., 2019*). Yet, research on the correlation between perceived stress and work engagement comparatively has been under examined.

## The mediating role of psychological resilience

Existing literature has found that positive psychological resources (*e.g.*, psychological resilience) can help individuals control and adapt to adjoining environments, improving their work engagement (*Corso-de Zúniga et al., 2020*; *Kašparkova et al., 2018*; *Mache et al., 2014*). Researchers have shown a correlation between poor psychological resilience and an exceeding level of work disengagement (*Villavicencio-Ayub, Jurado-Cárdenas & Valencia-Cruz, 2014*). Psychological resilience reflects people's adaptation process and positive attitude in the face of negative events (such as stress, trauma, adversity, frustration), which is characterized by vigor (high level of energy and physical activation), dedication (feeling of pride and enthusiasm with one's work), and absorption (being happily immersed in the work; *Schaufeli et al., 2002*).

The integrated psychological resilience model proposed by *Kumpfer (1999)* emphasizes the effect of selective perception, cognitive reorganization and other processes on

psychological resilience. Personal cognitive processes of events, such as the cognition and evaluation of stressful events, can affect psychological resilience. In a similar vein, considering both the internal and external contexts, *Mandleco & Peery (2007)* have suggested the system model theory, which emphasizes cognitive tendency (such as interpretation and perception of stressful events) is an internal influencing factor, which could have a great impact on psychological resilience. Previous studies have also confirmed that perceived stress negatively affects college students' psychological resilience—where elevated perceived stress denotes a comparatively lower level of psychological resilience (*Liu & Wang, 2016*; *Khosravi & Nikmanesh, 2014*). Therefore, perceived stress is likely to reduce the work engagement of volunteers by reducing their psychological resilience.

## Characteristics of variables in volunteers at different ages

Previous studies have divided adulthood into three different stages: early (18–35 years old), middle (35-60 years old) and late (post-60 years old; *Baikeli et al., 2021*). In China, a large number of volunteers are aged between 18 and 60 years. With economic development and social changes, although the three different stages of adulthood are not separated clearly, it should be noted that different developmental tasks are completed at different stages of adulthood. Specifically, with increased age, people's cognition of difficult situations and pressure as well as their attitude towards work could have some corresponding changes (*Almira et al., 2015*; *Wille et al., 2014*). For early adults, they may lack the ability to solve some problems and have low social experience, resulting in a lack of positive psychological qualities when dealing with problems (*Hershey & Farrell, 1999*). However, with age, problem-solving skills and abilities improve and more complex and difficult situations can be addressed more actively—often showing more resilience and optimism in the face of setbacks (*Lipsitt & Demick, 2012*). As a result, resilience may also increase with age.

According to the job demands-resources (JD-R) model, the influencing factors of work engagement are divided into demands and resources factors (*Bakker & Demerouti, 2017*; *Schaufeli, 2003*). However, this theory only emphasizes the significance of job characteristics and situational considerations of work engagement, ignoring the key role of individual cognitive factors (*Li et al., 2014*). With age, cognitive factors play a more substantial role than external situational factors for adults (*Ramos, Gregor & Georg, 2016*). Therefore, older adults' work engagement may be easily influenced by internal factors. It has been found that there are differences in resilience at different stages of adulthood (*Smith & Hayslip, 2012*). Generally, the resilience in early adulthood is not mature, and it is difficult to cope well with stressful events. Moreover, due to the lack of social experience, setbacks tend to be treated with subjective experience. Compared with the early adults, those in middle adulthood tend to view difficulties more positively. Even if they often encounter failures in the process, those in middle adulthood are more likely to persevere rather than give up easily. The resilience of the middle adults is relatively mature and more flexible in dealing with complex problem behaviors (*Riehm et al., 2020*). Consequently, there might be an age-dependent variation in the mediating influence of resilience on perceived stress and work engagement. Compared to the relatively mature mid-adults, the psychological

resilience of the early adults is more likely to mediate between perceived stress and work engagement.

## The present study

In sum, this study examined whether resilience during the outbreak of COVID-19 pandemic mediated the relationship between perceived stress and work engagement among volunteers. Based on existing literature, we predicted that perceived stress negatively impacts work engagement only through resilience, and the mediating effect of resilience was significant in volunteers in early adulthood.

## MATERIALS & METHODS

### Participants and procedure

In early November 2021, 981 volunteers ($M_{age}$ >18) were recruited online from an area in Gansu Province, China, where the first large-scale COVID-19 outbreak occurred in November 2021. Community volunteers included those working on the frontlines of the pandemic (excluding those in healthcare); students with medical and psychological backgrounds, teachers, civil servants, and freelancers formed the volunteer community. Community volunteers were mainly responsible for order maintenance, food distribution, emotional relief, and other work. Because of the severity of the COVID-19 pandemic at the time of the study, the volunteers recruited were taking on a heavy workload, with at least four hours of volunteer work a day until the pandemic was over. In order to reduce direct contact during the COVID-19 pandemic, we adopted the online questionnaire distribution method. We set up an online group for the subjects and distributed the link of the online self-assessment questionnaire to the recruited subjects at the same time. Before initiating the online self-reporting survey, mandatory informed consent was collected from each participant. With a 92.8% effective response rate, 910 valid completed questionnaires were compiled. The average age of the respondents was 37.45 ($SD = 3.16$) years, with 402 male and 508 female participants. In this study, the age of volunteers was divided into early adulthood and middle adulthood, corresponding to 18–35 years old and 36-60 years old, respectively (*Baikeli et al., 2021*). There were 503 in early adulthood, 260 male and 243 female corresponding average age of 26.03 ($SD = 4.81$) years, and 407 in middle adulthood, 195 male and 212 female with an average age of 40.36 ($SD = 3.01$) years. All the volunteers who participated in this study worked on the front lines of epidemic prevention work for nearly a month, and none reported a positive COVID-19 test result. We have received the permission to use the Chinese version of the Connor-Davidson Resilience Scale (C-CDRISC) and the Chinese version of the Utrecht Work Engagement Scale (C-UWES) from the copyright holders duly for this present research purpose. This present study completely complied with the Code of Ethics of the World Medical Association (Declaration of Helsinki) and its later amendments. Furthermore, the Ethical Board of Northwest Normal University has approved this study with ERB Number 20210256, dated: 01/11/2021. Participants' demographic characteristics are presented in Table 1.

**Table 1 Demographic information of the participants.**

| Variables | Groups | Frequency (%) |
|---|---|---|
| Gender | Female | 508 (55.8%) |
| | Male | 402 (44.2%) |
| Age | 18–35 | 503 (55.3%) |
| | 36–60 | 407 (44.7%) |
| Level of education | Junior high school and below | 131 (14.4%) |
| | High school | 233 (25.6%) |
| | Some College | 448 (49.2%) |
| | Bachelor's degree or above | 98 (10.8%) |
| Occupation | Student | 267 (29.3%) |
| | Teacher | 212 (23.3%) |
| | Civil servant | 168 (18.5%) |
| | Freelancer | 263 (28.9%) |

## Measures

### Perceived Stress Scale (PSS-10)

The Perceived Stress Scale (PSS) is a self-report questionnaire used to evaluate an individual's stress levels during the preceding month (*Cohen & Wills, 1985*). The current applied perceived stress scale has three variants of 14 items (PSS-14), 10 items (PSS-10) and four items (PSS-4) (*Ezzati et al., 2014*). Over the years, the scale has demonstrated good reliability and validity in multiple countries and populations (*Katsarou et al., 2012*; *Remor, 2006*; *Ruisoto et al., 2020*). The 10-item version was used in this study, which included 6 items (items 1, 2, 3, 6, 9, 10) of forward scoring to measure crisis perception factors; and four reverse scored items to measure perceived coping factors (items 4, 5, 7, 8). The sum of the two factor scores is the total score. Each item was scored on a 5-point Likert scale (0 = *never* to 4 = *very often*), with the total score ranging from 0 to 40. A higher total score of PSS-10 denotes the respondents' perceived stress level. For this present study, PSS-10 had good internal consistency reliability (Cronbach's $\alpha = .88$). A good structural validity of the PSS-10 for the single factor structure was demonstrated by a confirmatory factor analysis ($\chi^2 = 66.24$, $df = 24$, Comparative Fit Index [CFI] = 0.91, Tucker-Lewis Index [TLI] = 0.93, Root Mean Square Error of Approximation [RMSEA] = 0.04, Standardized Root Mean Square Residual [SRMR] = 0.04).

### Chinese version of Connor-Davidson Resilience Scale (C-CDRISC)

To access resilience, this present study administered revised the Chinese version of the Resilience Scale originally compiled by *Connor & Davidson (2003)* and revised by *Yu & Zhang (2007)*. The revised version contains 25 items and three dimensions of tenacity, self-reliance and optimism consistent with Chinese characteristics. Participants have responded on a 7-point Likert scale (1 = *no confidence at all* to 7 = *have full confidence*). This higher total score denotes participants elevated level of the resilience trait. The internal consistency in the present study was appropriate (Cronbach's $\alpha = .91$). A good structural validity of the C-CDRISC for the single factor structure was demonstrated by the confirmatory factor analysis ($\chi^2 = 92.82$, $df = 26$, CFI = 0.92, TLI = 0.93, RMSEA = 0.07, SRMR = 0.05).

***Chinese version of Utrecht Work Engagement Scale (C-UWES)***
To access the work engagement, this present study administered revised C-UWES originally compiled by *Schaufeli et al. (2002)* and revised by *Zhang & Gan (2005)*. The revised version comprises 15 questions to access three dimensions: vitality, dedication, and concentration. Participants have responded on a 7-point Likert scale (1 = *never* to 7 = *always*). The higher total score of C-UWES denotes the participant's higher level of work engagement. For this present study, C-UWES had good internal consistency reliability (Cronbach's $\alpha$ = .92). A good structural validity of the C-CDRISC for the single factor structure was demonstrated by the confirmatory factor analysis ($\chi^2$ = 78.47, *df* = 19, CFI = 0.94, TLI = 0.92, RMSEA = 0.04, SRMR = 0.05).

## Statistical analysis
For this present study, IBM SPSS v26.0 software was used for descriptive statistical analysis, including percentages, means, standard deviations, T-tests, and correlational analyses. For the purpose of hypothesis testing, IBM AMOS v25.0 was used to construct the models and compare structural equation models in multiple groups (*Arbuckle, 2009*). Power analysis was run using a priori sample size calculator for structural equation models. The required sample size was 323, with 80% power (*Soper, 2020*). Selected CFI, TLI, RMSEA, SRMR, Chi-square ($\chi^2$), and degrees of freedom (*df*) as the fitting index of the model, which CFI > 0.09, TLI > 0.09, RMSEA<0.08, SRMR<0.05 referred to a reasonable fit (*Browne & Cudeck, 1993*; *Kline, 2005*).

# RESULTS

## Characteristics and relationship of study variables in volunteers at different ages
### Characteristics of variables in volunteers of different ages
The age of volunteers (early adulthood *vs.* middle adulthood) was taken as the independent variable and perceived stress, resilience and work engagement were taken as the dependent variables for independent samples T-tests. The study results exhibited that there was nonsignificant difference between early and middle adult volunteers on perceived stress ($p$ > .577). The psychological resilience of early adult volunteers was comparatively lower among the middle adult volunteers, $t(908) = -13.30$, $p < .001$, $d = 0.88$. The work engagement level of early adult volunteers was significantly lower than that of middle adult volunteers, $t(908) = -8.91$, $p < .001$, $d = 0.60$. Table 2 demonstrates mean and standard deviation for all threes study variables.

### Correlation analysis of the main variables in volunteers at different ages
Descriptive statistics and correlation analyses of the main variables are shown in Table 3. Overall and at different age stages, perceived stress was significantly negatively correlated with resilience and work engagement, while resilience was significantly positively correlated with work engagement.

**Table 2  Descriptive statistics of volunteers' perceived stress, resilience and work engagement ($N = 910$).**

| Variable | Early adulthood ($n = 503$) M (SD) | Middle adulthood ($n = 407$) M (SD) | Cohen's d | Full Sample M (SD) |
|---|---|---|---|---|
| Perceived stress | 25.57 (4.02) | 25.31 (4.64) | | 25.55 (4.99) |
| Resilience | 81.72 (4.76) | 85.80 (4.47) | 0.88 | 82.72 (5.09) |
| Work engagement | 51.51 (6.79) | 55.36 (5.93) | 0.60 | 51.85 (6.45) |

Notes.

M, mean; SD, standard deviation.

**Table 3  Correlation analysis of perceived stress, resilience and work engagement.**

| Group | Variable | Perceived stress | Resilience |
|---|---|---|---|
| Early adulthood | Perceived stress | – | |
| | Resilience | $-0.33^{**}$ | – |
| | Work engagement | $-0.23^{**}$ | $0.52^{**}$ |
| Middle adulthood | Perceived stress | – | |
| | Resilience | $-0.67^{***}$ | – |
| | Work engagement | $-0.12^{*}$ | $0.10^{*}$ |
| Full sample | Perceived stress | – | |
| | Resilience | $-0.55^{***}$ | – |
| | Work engagement | $-0.18^{**}$ | $0.54^{***}$ |

Notes.

$^{*}p < .05$
$^{**}p < .01$
$^{***}p < .001$

### Mediating effect of resilience

Structural equation model (SEM) was used to investigate the mediating effect of volunteer resilience on perceived stress and work engagement. Perceived stress was an exogenous latent variable, and work engagement and resilience were endogenous latent variables (Fig. 1). As latent variables, perceived stress included crisis perception (a1) and perceived coping (a2). The latent variables of resilience included tenacity (b1), self-reliance (b2) and optimism (b3). The latent variables of work engagement include vitality(c1), dedication (c2), and concentration (c3). The results showed that the fitting indexes of the mediation model were $\chi^2 = 57.97$, $df = 17$, CFI = 0.98, TLI = 0.98, RMSEA = 0.05, SRMR = 0.03 and the fitting indexes were all satisfactory.

The path coefficients and relationships among the three variables can be seen from Fig. 1. The perceived stress of volunteers shows a significant direct negative predictive effect on resilience and work engagement, while resilience has a comparatively significant positive predictive effect on work engagement. This suggests that resilience mediates the relationship between perceived stress and work engagement.

### Differences in the mediating effect of resilience for volunteers at different ages

Volunteers aged 18 to 60 years were divided into two groups, early and mid-adulthood. According to previous study findings, there are significant distinctions in perceived stress,

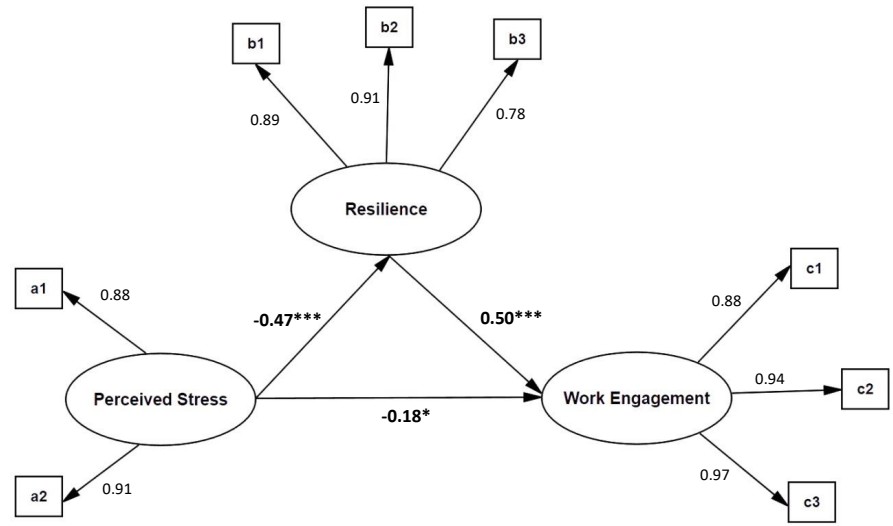

**Figure 1** **Mediating model of resilience on the relationship between perceived stress and work engagement.** The path coefficients are standardized. *$p < .05$, **$p < .01$, ***$p < .001$.

resilience and work engagement in different age groups. In order to determine whether the mediating effect of resilience is structurally consistent across ages, this study examined the mediating models of early and middle adulthood. The results showed that the indexes of early adult model were $\chi^2 = 69.71$, $df = 17$, CFI $= 0.98$, TLI $= 0.96$, RMSEA $= 0.06$, SRMR $= 0.04$. The indexes of the model were $\chi^2 = 60.02$, $df = 17$, CFI $= 0.98$, TLI $= 0.97$, RMSEA $= 0.04$, SRMR $= 0.03$. In general, all the fitting indexes are in an acceptable range and can be compared across groups (*Wen et al., 2004*). On this basis, this study adopts the method of multi-group comparison in SEM to set the equivalent model, and the fitting results of each model are demonstrated in Table 4. The result shows that the model between M1 and M2 was $\Delta\chi^2/ df = 2.74$, $p = .000$; the model between M2 and M3 was $\Delta\chi^2/ df = 2.50$, $p = .000$; the model between was M3 and M4 $\Delta\chi^2/ df = 3.68$, $p = .000$; and the model between M4 and M5 was $\Delta\chi^2/ df = 2.17$, $p = .000$. The difference of fitting index $\Delta$TLI and $\Delta$CFI between the two models is less than 0.01. This indicates that each equivalent model is valid (*Cheung & Rensvold, 2002*), suggesting the mediating model of resilience has the same significance and underlying structure in early and mid-adult volunteers.

Comparing the results of structural equation models for the different age groups, it was found that there were significant age differences in the impact of resilience on work engagement. The standardized path coefficients for the mediating models for the different age groups are shown in Fig. 2. For middle adult volunteers, the direct path coefficient of resilience to work engagement was not significant ($\beta = 0.06$, $p = .132$), and the direct path coefficients of perceived stress to work engagement were not significant ($\beta = -0.02$, $p = .095$). However, the path coefficient of perceived stress to resilience was significant ($\beta = -0.59$, $p = .000$). This suggests that resilience does not mediate between perceived stress and work engagement in mid-adult volunteers.

**Table 4   Three groups of equivalence fitting indexes of the mediation model.**

| Model | $\chi^2$ | df | CFI | TLI | SRMR |
| --- | --- | --- | --- | --- | --- |
| M1 | 160.78 | 39 | 0.98 | 0.97 | 0.02 |
| M2 | 168.99 | 42 | 0.97 | 0.96 | 0.03 |
| M3 | 171.49 | 43 | 0.97 | 0.96 | 0.03 |
| M4 | 178.85 | 45 | 0.97 | 0.96 | 0.03 |
| M5 | 183.19 | 47 | 0.97 | 0.97 | 0.03 |

Notes.

M1 is the measurement coefficient equality model; The structural coefficient equality model is added on the basis of M2 as M1. The structural covariance equality model was added on the basis of M3 as M2. On the basis of M4 is M3, the structural residual equality model is added; M5 is a measurement residual model based on M4.

df, degrees of freedom; CFI, Comparative Fit Index; TLI, Tucker-Lewis Index; SRMR, Standardized Root Mean Square Residual.

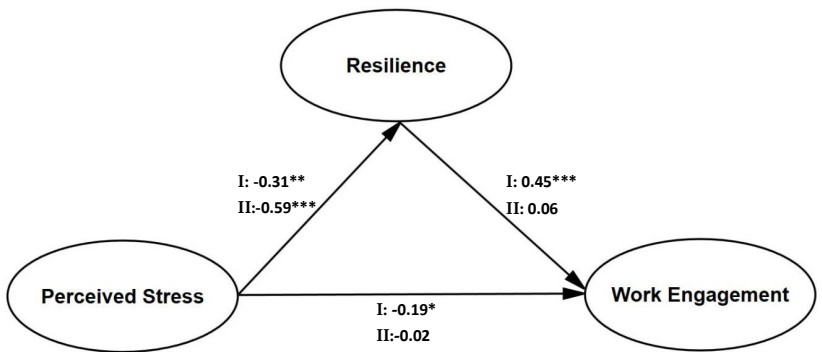

**Figure 2   Age differences for the mediation of resilience on the relationship between perceived stress and work engagement.** I: early adulthood; II: middle adulthood. The path coefficients are standardized. $^*p < .05$, $^{**}p < .01$, $^{***}p < .001$.

## DISCUSSION

This present study tested the mediation model to interpret how perceived stress influenced the level of work engagement in anti-epidemic volunteers while the COVID-19 outbreak. More specifically, the present study explored whether perceived stress affected volunteer work engagement for different ages and identified whether an underlying mechanism (*i.e.,* psychological resilience) impacted volunteers' work engagement for those age groups.

Findings of this study showed that volunteer work engagement increased significantly beginning in mid-adulthood. This confirmed a previous finding, people were more engaged at work after age 35 (*Suomäki, Kianto & Vanhala, 2019*). Because of age, volunteers in mid-adulthood have more social experience and can deal with difficulties in work more effectively, which greatly increases their level of psychological resilience. Furthermore, we found an interesting correlation between work engagement and volunteers of different ages. Compared with early adulthood volunteers, who may be more vulnerable to external influences, middle adulthood volunteers showed a higher level of work engagement—which may be due to a more stable developmental age and internal drive. The results may indicate the complexity of development on engagement in tasks for volunteers in early adulthood.

For those in early adulthood, who have been in society for a shorter time than middle adults, they may feel more compelled to prove their abilities to meet societal expectations. Early adults may also have a strong desire to pursue their inner selves to realize their self-worth. Thus, for early adult volunteers, what defines good circumstances for work engagement may vary depending on their balance of internal and external influences.

Additionally, the results demonstrated that perceived stress could predict volunteers' work engagement during the COVID-19 epidemic, which is homogeneous consistent with the results of previous studies previous studies findings (*Corso-de Zúniga et al., 2020*; *Wilks & Croom, 2008*). Several studies showed that elevated levels of perceived stress are associated with an upsurge in negative emotion like anxiety and depression (*Deo et al., 2020*; *Judit et al., 2017*). High perceived stress will not only have an adverse effect on volunteers' mental health, but also negatively impact their working state. In essence, when people perceive a high degree of stress, it could lead to a decrease in their positive emotion or even avoidance to positive information (*Khodarahimi, Hashim & Mohd-Zaharim, 2012*). Despite the knowing that positive emotions are beneficial, under high perceived stress people will react more greatly to the stress rather than fostering positive emotions. If this pattern extended to work, people may feel fear, be avoidant or even unwilling to concentrate on work under high perceived pressure. However, people with lower perceived stress levels usually report less negative emotion, respond to work in a positive coping style, and are more engaged in work (*Zhang et al., 2021*). The finding of perceived stress among volunteers working on the frontlines of the COVID-19 increasing their negative emotion would seriously affect their ability to engage in work is supported. In this study, no difference was found in the perceived stress between the two groups, which may be because during the outbreak of the epidemic, volunteers of different age groups were all under great psychological pressure and volunteer work pressure. Further it could be that those who were volunteers did not appraise COVID-19 as a threat/stressful compared to those who were not volunteers, which could be why there were no differences with age among volunteers. In the face of short-term high-intensity pressure every day, the perceived stress of the two groups was generally high and showed no significant difference.

This cross-sectional study supports that psychological resilience mediates the relationship between perceived stress and work engagement. Our results suggest that although perceived stress can negatively predict work engagement, resilience can reduce the adverse consequences of perceived stress on work engagement. This is homogenous with the findings of the previous study (*Yan et al., 2021*), which elucidated that it is difficult for groups with high perceived stress to deal with their problems in a positive and effective way. There are three core characteristics of psychological resilience: tenacity, self-reliance, and optimism. Tenacity is when a person does not simply give up but perseveres in the face of obstacles or loss of goal-directed behavior; self-reliance is the ability of an individual to recover and grow stronger after a trauma, where optimism refers to being confident and strong enough to try something new (*Connor & Davidson, 2003*). These positive psychological qualities can make people willing to put energy into working and living, while also helping them have adversity when there is pressure or problems at work (*Hakanen & Lindbohm, 2008*; *Chaudhary, 2020*). A positive working state, or high work

engagement, mainly reflects that individuals have an abundant energy at work and a strong tenacity when encountering difficulties. It is important that people have a healthy outlook and can skillfully adapt when faced with challenges at work. One way to do this would be through a dispositional quality, such as resilience, to help work toward living in harmony with the environment and having a positive attitude. In a few studies, resilience is a psychological trait that people need to adapt to complex situations and has played an important role in healthy development (*Yi et al., 2010*). Volunteers working during the COVID-19 pandemic may have faced high-pressure environments and thus may adjust their work to cope with the situations. If volunteers had psychological traits, such as resilience, they would be better engaged in their work.

Multigroup analysis of volunteers at different stages showed the differences in resilience mediated the relationship between perceived stress and work engagement in early adulthood, but the mediating effect was not significant in middle adulthood. The current results may have two possible reasons. First, differences in social development tasks in adulthood are a potential explanation. Early adulthood has a greater emphasis on accepting social responsibility to meet social expectations, while middle adulthood regards accepting social responsibility to meet self-worth; middle adulthood volunteers' social development could be seen as more stable and mature (*Hutteman et al., 2014*). Second, different levels of resilience are another possible reason. Development of three dimensions of resilience in middle adulthood tended to be complete, but in early adulthood only a single dimension was found (*Ong, Bergeman & Boker, 2009*). With the emergence of the COVID-19 variants, contamination become more challenging and volunteers were working in a more risky and complex situations. Volunteers in early adulthood should work to improve their resilience levels and adapt to situations more smoothly, so that engagement in work is consistent.

Yet, in terms of resilience, we do not have data to directly compare how this sample was before the study; thus, to understand the resilience results we draw upon prior literature. In the absence of COVID-19, earlier studies have found that early adults may lack the ability to solve some problems and have low social experience, resulting in a lack of positive psychological qualities when dealing with problems (*Hershey & Farrell, 1999*). However, with age, problem-solving skills and abilities improve and more complex and difficult situations can be addressed more actively, with older adults often showing more resilience and optimism in the face of setbacks (*Lipsitt & Demick, 2012*). As a result, resilience also increases with age. This is supported by the results of this study, which found that volunteers in mid-adulthood showed higher levels of resilience than those in early adulthood even after the COVID-19 outbreak, suggesting that age does affect resilience.

## Study strengths, limitations, and future research scope

The findings of this present study may have practical as well as theoretical implications. Apparently, it is inescapable for volunteers to perceive elevated levels of stress during the COVID-19 outbreak. During this unprecedented moment, the findings of this present study provide insight into the particular pathways through which perceived stress results in decreased work engagement of volunteers. Stress at work is inevitable, but our research suggests that increased resilience might help keep workers engaged regardless of the stress.

Hence, it is necessary to look into a practical ancillary means for volunteers to build resilience and, if at all conceivable, shift focus to volunteers to implement a successful intervention.

Previous studies without COVID-19 have found that perceived stress does affect the work engagement of different workers (*Abbas et al., 2021*; *Adanaqué-Bravo et al., 2023*; *Cheng & Kao, 2022*; *Kumar et al., 2021*; *Romero-Martín et al., 2022*; *Saleem, Malik & Qureshi, 2021*; *Zhang et al., 2021*). For this study, we selected workers who were most responsible for controlling COVID-19 during the pandemic. When stay at home orders were in place, volunteers made sacrifices to their personal safety to ensure their communities did not spread COVID-19. Although the sample pertained to COVID-19 volunteers, the volunteers included in the study came from a variety of backgrounds; therefore, it is plausible that these findings could be extended beyond a COVID-19 context. The results of this study support those of previous studies in terms of how perceived stress and resilience impacts work engagement. This study adds a new contribution to the literature by identifying a mechanism of perceived stress that affects work engagement of volunteers across different age groups during COVID-19. Future work should continue to explore how volunteers' work engagement is impacted as well as identifying other mechanisms or moderating factors that could impact this relationship. Overall, the results of this study not only work to fill a gap in the literature but can also be useful for informing interventions to increase work engagement for volunteers so that the prevention and control of COVID-19 can be better maintained and monitored. The results of this study could also be informative for future public health emergencies, where the government may similarly establish a volunteer team.

This present study is attributed with some inevitable limitations. Self-reported survey questionnaires measured the research variables. In contrast to the thorough analysis involved in a qualitative method, certain important information was likely left out of a quantitative assessment. Furthermore, we recruited Chinese samples exclusively, which undoubtedly thwarted generalizability. Hence, it is necessary for future research to investigate and study volunteers in epidemic areas in other countries. Eventually, the substantial direct impact between perceived stress and work engagement recommends that other mediators or moderators might not have comprised through this study. Considering that the COVID-19 epidemic may seem unrestrained for some people, the potential to regulate negative feelings in response to stressful circumstances and adopting emotional regulation tools should be considered as a moderator. Future research might benefit from taking a broader view of the identified components in order to fill in the gaps in our existing findings of the connections between them.

## CONCLUSIONS

In conclusion, the findings of this study suggest that work engagement increased with age and was negatively predicted by perceived stress, showing these factors were important for volunteers' work during COVID-19. Further, for volunteers during COVID-19 in early adulthood, psychological resilience mediated the relationship between perceived stress

and job engagement. Compared with previous research, this study could have important implications for understanding the mechanisms through which perceived stress impacts job engagement.

### Funding
The authors received no funding for this work.

### Competing Interests
The authors declare there are no competing interests.

### Author Contributions
- Yatong Li conceived and designed the experiments, performed the experiments, prepared figures and/or tables, and approved the final draft.
- Pei Xie performed the experiments, prepared figures and/or tables, and approved the final draft.
- Liang He analyzed the data, authored or reviewed drafts of the article, and approved the final draft.
- Xiaolong Fu analyzed the data, authored or reviewed drafts of the article, and approved the final draft.
- Xiaobing Ding analyzed the data, authored or reviewed drafts of the article, and approved the final draft.
- Mary C. Jobe analyzed the data, authored or reviewed drafts of the article, and approved the final draft.
- Md Zahir Ahmed analyzed the data, authored or reviewed drafts of the article, and approved the final draft.

### Human Ethics
The following information was supplied relating to ethical approvals (*i.e.*, approving body and any reference numbers):

Ethics Board of Northwest Normal University.

### Data Availability
The raw data are available in the Supplemental Files.

### Supplemental Information
Supplemental information for this article can be found online at http://dx.doi.org/10.7717/peerj.15704#supplemental-information.

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
