# Peer review of "The effect of perceived stress for work engagement in volunteers during the COVID-19 pandemic: the mediating role of psychological resilience and age differences"

_PeerJ, doi:10.7717/peerj.15704_

## Round 0.1 · original submission · Major Revisions

I have now received the reviewers' comments on your manuscript. They have suggested some revisions to your manuscript. Therefore, I invite you to respond to the reviewers' comments and revise your manuscript.

Reviewer 1 ·

Basic reporting

The paper deals with how perceived stress is correlated with work engagement in volunteers during the COVID-19 pandemic. This is the interesting topic to be analyzed and might draw an attention from related researchers. However, there are serious concerns of estimation method and its interpretation.

Experimental design

The most serious problem is the issue of causality. Authors attempted to investigate the causality by assuming that “Perceived stress was an exogenous latent variable, and work engagement and resilience were endogenous latent variables (Figure 1).” (Lines 257-258).

I am major in Economics and so provide comment on it from basic econometric point of view although authors are researchers in different field. I cannot agree with the assumption. It is most important issue to provide evidence of causality between dependent and independent variables (Angist and Pischke, 2008). To this end, exogeneity of independent variables should be strictly considered. For instance, estimation results suffer endogeneity bias if there is reverse causality. In this case, work engagement might improve mental condition, which improve perceived stress. That is, subjective variables as independent variables inevitably suffer endogenous bias. In empirical economic researcher developed various technique to deal with the problem such as Difference-in-difference method, regression discontinuity method apart from traditional Instrumental variable (Angist and Pischke, 2008).
Anyway, author hardly consider the point.

Angist, J., and Pischke, J. 2008. Mostly Harmless Econometrics: An Empiricist's Companion. PRINCETON UNIVERSITY PRESS

Validity of the findings

1)We found various arrows with degree of impact and statistical significance in Fig.1. However, as I criticize in the former comment, causality has not been scrutinized. This is probably because mediating model has not been acknowledged by economist like me. I cannot accept such the model automatically calculate the causality without any consideration of endogenous biases.

2) Authors did not control basic factors such as household income level, work status, and marital status. Therefore, omitted variables biases naturally occur.

3) Observations are 910, which is too small to obtain reliable results.

Reviewer 2 ·

Basic reporting

The study focuses on a fairly interesting question and uses data obtained in a realistic setting, during the first month of the COVID pandemic outbreak in the area. However, the presentation of the analyses and the findings is lacking important detail, making it difficult to understand what exactly was done by the authors at each step and to evaluate the conclusions; some analyses are also questionable. I believe that these issues warrant a major revision and another round of reviews.

- English needs editing with respect to usage, to make sure all the sentences are clearly comprehensible (e.g. Line 50 “the contributions of this volunteers-being…”, Line 72 “intimately correlated”), as well as spelling (e.g. Line 51 “extend” instead of “extent”).

- Please provide effect sizes (Cohen’s d) in Table 3.

- The obvious strength of the study is the fact that the data were collected in a very realistic setting, during the first month of the pandemic outbreak in the area, when the pandemic-related stress process was developing in the volunteers. It might be good to emphasize this in the abstract, as this fact somewhat mitigates the weaknesses related to the cross-sectional study design.

- Did the volunteers have regular jobs – were they laid off during the lockdown, or were they working from home and volunteering part time? This is important in order to understand whether the volunteers completed the UWES with respect to their volunteer work (during the pandemic) or were they free to understand the questions as pertaining to their regular job (out of the pandemic context)? Was the volunteering/pandemic context somehow specified in the UWES instructions?

- The power analysis is mentioned in Line 208, but no results are provided. The same goes for F tests (Line 205) – why mention them if the results are not presented?

- In the Measures section, CFA results are presented for each measure. However, it’s completely unclear what the measurement model was. There is also no chi-square value and no degrees of freedom, making it impossible to infer the measurement model. Was it a single-factor model based on all the items, or a bifactor model (e.g., for PSS with reverse-scored items), or a model based on some item parcels? Without this information, the fit indices are meaningless – please provide the information on the measurement models.

- Please do not use the chi-square to df ratio. Just as the chi-square statistic, this ratio is dependent on sample size, and it can give highly misleading results, especially for misspecified models. In fact, it is precisely in order to overcome these drawbacks that practical fit indices, such as the CFI and the RMSEA, were developed in the 1980s, rendering the X2/df ratio obsolete (see, e.g., West, S. G., Taylor, A. B., & Wu, W. (2012). Model fit and model selection in structural equation modeling. In: R. H. Hoyle (Ed.), Handbook of structural equation modelling (pp. 209-231). Guilford Press).

- The methodology of comparison between the nested models is not clear. The authors needs to mention explicitly, which parameters are held equal at each step – for example, which covariance is held equal in M3 (there is no covariance in the mediation model shown in Fig. 1!)? From a substantive point of view, there is probably no need to test the invariance of the residuals and covariances: it is enough to test the invariance of model paths. And it is here that the authors do something really strange, as the chi-squared differences reported in the text do not correspond to the values in Table 5. Based on Table 5, the chi-squared difference test indicates that the difference between M2 and M1 (168.99-60.78=8.21 with 3 df) is significant at p<.05. It would make sense to do Wald test for each model path separately to find out which path is significantly different (apparently, it’s the path from resilience to work engagement).

- The structural equation model is based on parcels. This strategy is acceptable to simplify the model, but it is a choice which requires some explanation of why parcelling was used and how exactly the parcels were created. This detail is needed, given that it is known that parcelling can conceal the problems with scale structure and that modelling results can depend on the way the items are grouped into parcels (Little, T. D., Rhemtulla, M., Gibson, K., & Schoemann, A. M. (2013). Why the items versus parcels controversy needn’t be one. Psychological methods, 18(3), 285-300).

Experimental design

- The test of common method bias makes no sense. Among the different types of bias described in the Podsakoff et al. (2003) paper, very few are going to influence the scores in the three measures in the same way. For instance, acquiescence bias is likely to affect the scores on the UWES and CD-RISC scores, but not the PSS, which has nearly a half of reverse-scored items. The social desirability bias is also going to affect the scores in different ways. It is not really possible to capture these biases by just modelling them as a single factor. What the factor would capture would be the common variance of the three measures, bias and non-bias included. Also, the authors mention that “after the common method factor was added into the three factors (Table 2), the fitting index of the model did not improve significantly” – this appears to be simply wrong, given the information in Table 2: the model is not described, but it is clearly nested within the three-factor model, and the chi-square difference test (57.97-29.07=28.90 for 17-9=8 degress of freedom, which is significant at p < .001) indicates that the fit of the model with a “method” factor is significantly better that that of a model without it. However, this is not really a “method” factor, because it would capture the variance of the direct effect and the indirect effect as well. There is no way to control for the common method bias the way the authors are trying to do without changing the design of the study and the measures, so it’s best to just remove it.

- I also have a strong reservation with respect to the theoretical model. The mediation model implies that resilience is reduced as a consequence of perceived stress and results in lower work engagement. While I subscribe to the idea that perceived stress and resilience are likely causes of work engagement, I am not confident at all concerning the place of resilience in this process proposed by the authors. While it is true that psychological resources underlying resilience can deplete as a consequence of stress as the authors note in the introduction, this process takes time and it can probably only be observed in a long term. In a short term, in line with Lazarus and Folkman’s model, resilience resources are more likely to act as a buffer, affecting the cognitive evaluations and reducing the impact of stressful events on psychological (perceived) stress, diminishing its adverse effects on work engagement. Thus, I would rather expect resilience, firstly, to predict lower perceived stress, and, secondly, to moderate its effects on work engagement (interaction stress x resilience). Thus, I would suggest to test an alternative mediated moderation model here. (Potentially, resilience could also moderate the effects of age on perceived stress and on work engagement: resilience is more important for older adults facing higher risks).

Validity of the findings

- The fact that perceived stress was not associated with age is really surprising, given that the COVID-related risks are age-dependent. Also, resilience is much higher in the older part of the sample (by nearly a standard deviation!) I would be very curious if there is any pre-pandemic data obtained for this CD-RISC adaptation with these age groups? If this pattern has existed before, it could be due to differences between age groups either in resilience or in response styles (such as acquiescence, given that the CD-RISC does not have any reverse-scored items), or both. If this difference did not exist in pre-pandemic scores (or was much weaker), an explanation could be selection bias: given that COVID risks are age-dependent, among middle adults it would be only the more resilient individuals who would choose to take these risks and go volunteering (whereas in the group of young adults, individuals with different resilience levels would be more likely represented). Some discussion of these effects is needed.

Reviewer 3 ·

Basic reporting

no comment

Experimental design

Although the materials and methods section is well structured, some clarifications are still needed:
- How was the questionnaire distributed among respondents?
- In how many medical units did they work as volunteers? Was it the same level of severity of COVID-19 pathology?
- Were the characteristics of volunteering assessed (number of hours/day, number of days per week, shift work)?
- Were other demographic data collected, e.g. occupation? Were they medical staff or medical students among the volunteers?

Validity of the findings

The study assesses perceived stress and work engagement in volunteers in the context of the COVID-19 pandemic. Has the impact of the COVID-19 pandemic been assessed in any way? Is the current pandemic a modulating factor or can the results be extrapolated to volunteering regardless of the field?

---

## Round 0.2 · accepted · Accept

Thank you very much for improving and revising your manuscript based on the comments and suggestions of the reviewers.